# NEURAL BAYESIAN FILTERING

## ABSTRACT

We present Neural Bayesian Filtering (NBF), an algorithm for maintaining distributions over hidden states, called beliefs, in partially observable systems. NBF is trained to find a good latent representation of the beliefs induced by a task. It maps beliefs to fixed-length embedding vectors, which condition generative models for sampling. During filtering, particle-style updates compute posteriors in this embedding space using incoming observations and the environment's dynamics. NBF combines the computational efficiency of classical filters with the expressiveness of deep generative models—tracking rapidly shifting, multimodal beliefs while mitigating the risk of *particle impoverishment*. We validate NBF in state estimation tasks in three partially observable environments.

## 1 INTRODUCTION

Belief state modeling, or computing posterior distributions over hidden states in partially observable systems, has numerous applications in sequential estimation and decision-making problems, including tracking autonomous robots and learning to play card games (Haug, 2012; Sokota et al., 2022; Barfoot, 2024). As an example, consider the problem of tracking an autonomous robot with an unknown starting position in a $d \times d$ grid (Figure 1). Suppose the agent's policy is known, and an observer sees that the agent moved a step without colliding into a wall. This information indicates how the observer should update their *beliefs* about the agent's position. Tracking these belief states can be challenging when they are either continuous or too large to enumerate (Solinas et al., 2023)—even when the agent's policy and the environment dynamics are known.

A common approach frames belief state modeling as a *Bayesian filtering* problem in which a posterior is maintained and updated with each new observation. Classical Bayesian filters, such as the Kalman Filter (Kalman, 1960) and its nonlinear variants (e.g., Extended and Unscented Kalman Filters (Sorenson, 1985; Julier & Uhlmann, 2004)), assume that the underlying distributions are unimodal and approximately Gaussian. While computationally efficient, this limits their applicability in settings that do not satisfy these assumptions. Particle filters alternatively approximate arbitrary target distributions through sets of weighted particles. However, in high-dimensional state spaces, they can require maintaining exponentially large sets of particles or risking *particle impoverishment*—a phenomenon where the set contains very few particles with significant weight (Doucet et al., 2009).

Advances in generative modeling have provided new methods for filtering in problems with complex, multimodal belief states. However, they approximate the full system dynamics (including agent policies) and update an internal representation of the beliefs with each observation. This is a significant limitation in applications where the policy or environment is known but changes, which happens naturally in some learning algorithms (Moravčík et al., 2017; Schmid et al., 2023).

In this paper, we propose **Neural Bayesian Filtering** (NBF), which models complex, multimodal belief states and updates posteriors efficiently for input policies and environments. Central to our approach is the idea that belief states in a given task form a parameterized set. Much like how mean and variance parameterize the family of Gaussians, a learned embedding vector specifies a particular belief state instance. This embedding can be computed exclusively using samples from the target belief state—making it specific to a given policy, environment, and observation sequence. Given a new observation, NBF updates the embedding to approximate the new posterior by generating, simulating, and then re-embedding particles. Effectively combining particle filtering and deep generative modeling, the algorithm maintains expressive approximations of complex, multimodal belief states. We validate NBF empirically in variants of *Gridworld*, the card game *Goofspiel*, and a continuous localization environment called *Triangulation*.

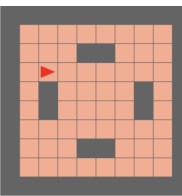 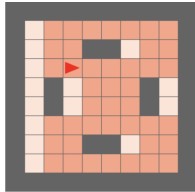 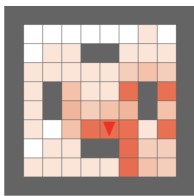

(a) 8x8 grid with initial beliefs.

(b) Beliefs after moving right.

(c) After more steps.

Figure 1: Tracking an agent with an unknown starting position from observations about which direction the agent moved (with some probability of error) and whether or not it hit a wall. Colored cells indicate probabilities of possible agent positions.

### 1.1 MAIN CONTRIBUTIONS

**Belief State Embeddings**  We propose learning an embedding network that compresses sample sets from belief states into a set-invariant vector. Conditioning a generative model on this vector allows for efficient sampling and density estimation on a family of complex posterior distributions.

**A Flexible Parametric Framework For Filtering**  We introduce Neural Bayesian Filtering (NBF), a novel parametric filtering framework that combines classical filtering, embeddings, and deep generative modeling. NBF can approximate multimodal, non-Gaussian, and discrete state distributions without prohibitively large particle sets or fixed parametric assumptions.

## 2 BACKGROUND

Belief state modeling has been studied in numerous contexts, including Hidden Markov Models (HMMs) (Rabiner, 1989), Partially Observable Markov Decision Processes (POMDPs) (Kaelbling et al., 1998), and Factored Observation Stochastic Games (FOSGs) (Kovařík et al., 2022), and is critical to many decision-time search algorithms. Sokota et al. (2022) provide a unified notation for belief state modeling. This work extends their formulation to sets of environments and explicitly models non-stationarity in the environment and control. These non-stationarities arise naturally when the agent learns or the environment changes (e.g. different obstacles in the grid in Figure 1).

### 2.1 NOTATION

Let $x \in X$ be a **Markov state** and $\pi \in \Pi$ be an external **control** variable (such as a policy in a POMDP or a joint policy in an FOSG). Let $T : X \times \Pi \to \Delta X$ be the **transition function** that determines the underlying dynamics. Emission function $H : X \times X \to \Delta Y$ outputs a probability distribution over the the **observations** (emissions) $y \in Y$ upon transition from $x$ to $x'$. An **environment** $G \stackrel{\text{def}}{=} (X, Y, T, H, p_0)$ contains state and observation spaces, a transition function, an observation function, and an initial state distribution $p_0$. In this paper, we extend this formulation to include a set of state spaces $\mathcal{X}$, transition functions $\mathcal{T}$, and observation spaces $\mathcal{Y}$ to model the scenario where an environment instance is known but non-stationary. $\mathcal{G} \stackrel{\text{def}}{=} \{(X, Y, T, H, p_0) : X \in \mathcal{X}, Y \in \mathcal{Y}, T \in \mathcal{T}, H \in \mathcal{H}, p_0 \in \Delta X\}$ is the set of environments.

Given an instance $(G, \pi)$, **belief state modeling** is expressed as modeling the distribution over the Markov states at a particular time $t$, conditional on the control and emission variables: $p(x|\pi, y^{(1)}, \ldots, y^{(t)})$. In discrete Markov systems with small state spaces, belief states can be computed analytically using posterior updates for each observation in the sequence.

In this work, we consider the problem where both $G$ and $\pi$ are known by the agent or external observer and computationally efficient to evaluate. Together, $\mathcal{G}$ and $\Pi$ parameterize a set of belief states $\mathcal{P}_G^\Pi \stackrel{\text{def}}{=} \{p(x|\pi, y^{(1)}, \ldots, y^{(t)}) : \pi \in \Pi, G \in \mathcal{G}, y^{(i)} \in Y, t \in \mathbb{N}\}$. For brevity, we will drop the conditional and refer to a member of this set as $p(x)$. Unlike much of the prior work in neural belief state modeling, our approach models this set directly by conditioning on specific $G$ and $\pi$.

## 2.2 Classical Filtering Algorithms

*Bayesian filtering* (Särkkä & Svensson, 2023) has been studied extensively as a method for belief state modeling. Environments with linear Gaussian dynamics are the simplest case. In these settings, **Kalman filters** (Kalman, 1960) provide efficient closed-form solutions for the posterior mean and covariance. However, many real-world applications involve nonlinear, non-Gaussian processes. Improvements such as Extended (Sorenson, 1985), Unscented (Julier & Uhlmann, 2004), and Cubature (Arasaratnam & Haykin, 2009) Kalman filters are suitable for non-linear systems but still propagate unimodal beliefs.

**Particle filters** (Doucet et al., 2009) maintain a representative set of weighted samples for the belief state. This set gets updated according to the environment dynamics upon each new observation. Particle filters can, in principle, approximate a wide range of distributions, but they come with other challenges. Particle impoverishment happens when many particles in the set have little or no weight given the observation sequence and can be catastrophic because replacement particles can only be sampled by duplicating others in the set (Sokota et al., 2022). Computational efficiency can also become a concern in high-dimensional state spaces because accurate filtering generally requires maintaining an exponential number of particles (Thrun, 2002). Specialized approaches for mitigating impoverishment (Murphy & Russell, 2001; Orguner & Gustafsson, 2008; Park et al., 2009) often lack generality, relying on exploiting problem-specific structure, domain knowledge, or hand-tuned heuristics, and possess the same challenges with resampling.

In the next two sections, we describe NBF: a novel approach that models the set of belief states parameterized by $\Pi$ and $\mathcal{G}$ as a latent space of *belief state embeddings*. These embeddings condition a generative model for sampling and density estimation. Approximate posteriors are updated by sampling particles from the embedding, simulating these particles using the input $G \in \mathcal{G}$ and $\pi \in \Pi$, and then computing a weighted embedding with the result. Learned neural embeddings and fast posterior computation in the embedding space let NBF combine the computational efficiency of parametric approaches like Kalman filters with the flexibility of particle or model-free methods.

## 3 Embedding Belief States

In this section, we formalize our approach to modeling complex, multimodal belief states using learned neural embeddings. Our goal is to represent and efficiently sample from a given belief state induced by known but potentially changing control variables and environment dynamics.

Given a known control variable $\pi \in \Pi$ and environment $G \in \mathcal{G}$, the induced belief state after $t$ observations is the posterior distribution $p(x)$. One challenge is efficiently modeling the set of these posterior distributions, which may be diverse as $\pi$ or $G$ vary. We propose embedding these distributions from sets of ground truth samples of example belief states. Our approach aims to construct an embedding vector $\theta \in \mathbb{R}^m$ that uniquely represents the posterior distribution defined by $y^{(1:t)}$, $\pi$, and $G$, and conditions a model for sampling and density estimation.

### 3.1 Model Definition

We model the target set of belief states using an embedding function and a Normalizing Flow (Papamakarios et al., 2021) conditioned on its output.

**Embedding Function.** Let $x_{1:n} \stackrel{\text{def}}{=} (x_1, x_2, \ldots, x_n)$ denote i.i.d. samples from belief state $p(x)$. We define a permutation and cardinality invariant function $\mathcal{E}_\phi : \mathcal{X}^n \times \mathbb{R}^n \to \mathbb{R}^m$ that maps (weighted) samples $x_{1:n}$ to an $m$-dimensional embedding vector. A **belief embedding** $\theta \stackrel{\text{def}}{=} \mathcal{E}_\phi(x_{1:n}, w_{1:n})$ approximates the salient features (e.g. shape, location, spread) of a target distribution $p(x)$ as a vector in latent space. Together with the flow described next, it defines the distribution $p_\theta(x)$.

Permutation and cardinality invariance ensures that neither $n$ nor sample order affects $\theta$. In this paper, we take the (weighted) mean over individual sample embeddings. If $\phi$ is expressive enough, the mean-pooled embedding $\theta$ serves as a sufficient moment-based approximation of $p(x)$, but other higher-order architectures such as DeepSets (Zaheer et al., 2017) may also be viable.

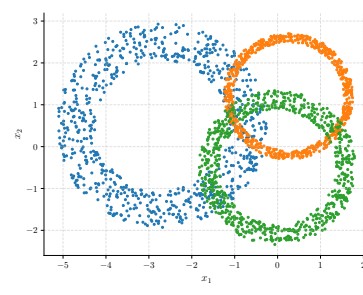 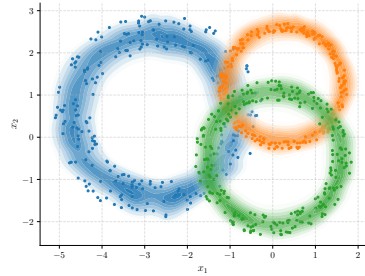

(a) Sample donut distributions. Each distribution has three parameters: mean, radius, and width.

(b) Learned densities after conditioning the model on 128 samples from the target distribution.

Figure 2: Embedding the set of donut distributions in $\mathbb{R}^2$

**Conditional Normalizing Flow.** Normalizing flows are a class of generative models that transform a simple base distribution (e.g., Gaussian) into a complex target distribution through a series of invertible and differentiable mappings. They allow for exact likelihood computation via change-of-variables. Flows can also be constructed in continuous time by defining an ordinary differential equation that describes the dynamics of the transformation over time (Lipman et al., 2022).

Given an embedding $\theta$, we can define tractable sampling and density estimation operations on $p_\theta(x)$ by conditioning a normalizing flow on $\theta$. Let $f_\psi(\cdot; \theta) : \mathbb{R}^d \to \mathbb{R}^d$ be an invertible, differentiable transformation conditioned on $\theta$ and parameterized by $\psi$. Given a simple base distribution $p(z)$ (e.g. standard normal), the following two-step sampling procedure:

$$z \sim p(z); \ x = f_\psi(z; \theta),$$

gives the desired density $p_\theta(x)$ (by change-of-variables) (Papamakarios et al., 2021):

$$p_\theta(x) \overset{\text{def}}{=} p\left(f_\psi^{-1}(x; \theta)\right) \left| \det \frac{\partial f_\psi^{-1}(x; \theta)}{\partial x} \right|.$$

If $f_\psi(z; \theta)$ has a tractable inverse, then evaluating this density is also tractable. $\psi$ and $\phi$ can be optimized jointly by maximizing the log-likelihood over all samples $1, \ldots, N$ and distributions $1, \ldots, K$ in the training set:

$$\mathcal{L}(\phi, \psi) \overset{\text{def}}{=} \sum_{k=1}^{K} \sum_{i=1}^{N} \log p_\theta^{(k)}(x_i)$$

**Discrete Belief States and Variational Dequantization.** Normalizing flows are defined for continuous inputs, but belief states in many relevant domains are discrete. *Variational Dequantization* (Ho et al., 2019) is an approach for applying flows to discrete data. Each discrete sample is perturbed by learned noise, resulting in a continuous space. The noise distribution is trained jointly with the flow by maximizing a variational lower bound on the true discrete log-likelihood—preserving exact likelihood evaluation and stabilizing training for discrete belief states.

## 3.2 ILLUSTRATIVE EXAMPLE: DONUTS

Consider a domain, which we call donuts, consisting of a simple set of continuous distributions in $\mathbb{R}^2$. Donuts (Figure 2(a)) are parameterized by a mean, a radius, and a width. Setting these parameters specifies a particular donut $D$, and they are sufficient for closed-form sampling from $D$. Suppose these parameters are not known, and instead $\mathcal{E}_\phi$ receives a set of sample points $x_{1:n} \overset{\text{iid}}{\sim} D$ and outputs $\theta$ as a parameterization of $D$. $\theta$ conditions the generative model $f_\psi(z; \theta)$, providing an approximation of $D$ and enabling i.i.d sampling from it.

We train our model on randomly generated example donuts using $n = 128$ samples per donut—with $n/2$ used for generating the embedding and the rest used for minimizing the negative log-likelihood

objective. At test time, we create embeddings using 64 samples from unseen target donuts. Figure 2(b) shows 3 randomly selected test donuts. Points are samples from the target distribution used to generate $\theta$, while the contours are generated by evaluating the log densities of grid points according to $f_\psi^{-1}(x; \theta)$. Hyperparameters and other training details can be found in the appendix.

## 4 FILTERING WITH EMBEDDINGS

If $p(x)$ is a posterior induced by a sequence of observations, obtaining the samples needed to compute $\theta$ may carry significant computational overhead. More general-purpose belief state approximation using our model requires tracking the belief state over the observation sequence. In this section, we describe an algorithm that tracks belief states in the embedding space.

Upon receiving an observation, classical parametric methods, such as variants of Kalman filters, compute the posterior in closed form. If $\theta$ represents the parameters for a given belief state, such methods define an update function $g : \mathbb{R}^m \times Y \to \mathbb{R}^m$ such that $\theta' = g(\theta, y)$. The modeling assumptions that enable closed-form updates are often violated, which motivates approximate methods. Approximating $g$ for a fixed $G$ and $\pi$ is a viable choice, but often lacks efficient methods for parameterizing $g$ with $G$ and $\pi$ in settings where they are variable inputs.

Particle filters are non-parametric: they represent arbitrary target distributions as sets of weighted sampled points called particles. Posterior updates to these empirical distributions are performed by simulating transitions using $G$ and $\pi$ for each particle and updating weights according to the induced transition probabilities. Below we provide a typical posterior update for a particle filter given observation $y$, $\pi \in \Pi$, and $G \in \mathcal{G}$:

For each particle $x_i$ and particle weight $w_i$, $i \in 1, \ldots, n$:

1. Simulate transition $x_i' \sim T_G(x_i, \pi)$
2. Update weight $w_i' \leftarrow w_i \cdot H_G(x_i, x_i')[y]$

New weights are typically used to *resample* particles by duplicating particles that are more likely to match the updated observation sequence and discarding others. Impoverishment occurs when all particle weights become small and new particles cannot be resampled from outside $x_{1:n}$.

Neural Bayesian Filtering approximates belief states by performing a similar update in the embedding space of a pre-trained belief embedding model. It incorporates $\pi \in \Pi$ and $G \in \mathcal{G}$ into the posterior update like a particle filter, but avoids impoverishment by resampling from the model at every step. Given an embedding, NBF generates particles according to $p_\theta(x)$, simulates them forward while computing their weights exactly like a particle filter, and then computes a new weighted embedding from the result. Figure 3 shows an overview of NBF's posterior update. Full details, including pseudocode, are shown in the appendix.

### 4.1 CONVERGENCE OF NBF WITH A PERFECT MODEL

NBF is *consistent* and converges at the standard Monte-Carlo rate for finite $X$ and $Y$ under the following assumptions: (i) the embedding model is expressive enough to represent every belief state exactly, and (ii) there exists some global $\epsilon \in (0, 1]$ such that given an observation $y$, the probability of transitioning from any state $x$ to one of its successors $x'$ and observing $y$ is at least $\epsilon$. We call (ii) $\epsilon$-*global observation positivity* of $(G, \pi)$.

**Theorem 4.1** (NBF Consistency). *Assume $\epsilon$-global observation positivity of $(G, \pi)$ and a finite $X$ and $Y$. For any finite horizon $t_{max}$, belief state $p_t(x), t \leq t_{max}$, and any bounded function $\varphi : X \to \mathbb{R}$, let*

$$\hat{\mu}_t^{(n)}(\varphi) = \frac{\sum_{i=1}^n w_i \varphi(x_i)}{\sum_{i=1}^n w_i}$$

*be the estimate of $\mathbb{E}_{p_t}[\varphi]$ computed by NBF with a perfect embedding model and $n$ particles. Then,*

$$\sup_{0 \leq t \leq t_{max}} |\hat{\mu}_t^{(n)}(\varphi) - \mathbb{E}_{p_t}[\varphi]| \xrightarrow{a.s.} 0$$

*as $n \to \infty$.*

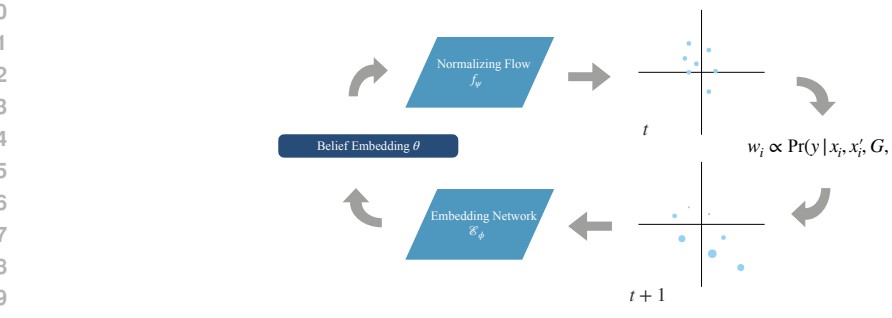

Figure 3: Neural Bayesian Filtering generates particles from a belief embedding, simulates them in the environment, and embeds them with a weight proportional to the probability of $y$.

Theorem 4.1 states that for any bounded function $f$ on the state space and belief state $p(x)$, NBF's estimate of $\mathbb{E}_p[\varphi]$ almost surely converges to the true value as its number of particles approaches $\infty$. The following Corollary states its convergence rate under the same conditions.

**Corollary 4.2** (NBF Convergence Rate). *Under the same conditions as Theorem 4.1, as $n \to \infty$,*

$$\sup_{0 \leq t \leq t_{max}} |\hat{\mu}_t^{(n)}(\varphi) - \mathbb{E}_{p_t}[\varphi]| = O_p(n^{-1/2})$$

Further details, including proofs, are included in the Appendix.

## 5 EXPERIMENTS

We validated Neural Bayesian Filtering in partially observable variants of Gridworld, the card game Goofspiel, and the localization environment *Triangulation*. Belief states in the first two environments are discrete, so we used variational dequantization as described in Section 3. Additional experiments and further details, including hyperparameter settings, are available in the supplementary material. Source code will be made available upon publication.

We compared a total of four approaches:

- **Approx Beliefs:** The embedding model described in Section 3 with access to $p(x)$ to generate samples for an embedding size of 32. Each training instance consists of 64 samples from some $p(x) \in \mathcal{P}_G^\Pi$. Model hyperparameters were not tuned extensively.

- **PF** $(n)$: A Sequential Importance Resampling Particle Filter with $n$ weighted particles representing the belief state. An effective sample size less than $n/2$ triggers a systematic resample (Doucet et al., 2009) of the particles.

- **NBF** $(n)$: A Neural Bayesian Filter with the same belief embedding model as "Approx Beliefs" and $n$ particles for posterior computation.

- **Recurrent:** A two-layer LSTM trained to predict $p(x) \in \mathcal{P}_G^\Pi$ from observations.

Performance was measured in terms of Jensen-Shannon (JS) divergence between the model's predicted belief state and the ground-truth posterior, with lower values indicating better performance.

### 5.1 PARTIALLY-OBSERVABLE GRIDWORLD

We conducted experiments on a partially observable variant of Gridworld with grids of size 5 and 8, and dimensionality 2 and 3. Each grid contains a fixed number of square (or cube) obstacles, and every agent step results in an observation indicating whether the agent hit a wall. For each dimensionality and size, we evaluated performance in two conditions: a *fixed* grid and policy and a *randomized* grid and policy, yielding eight total experimental configurations (5-2D-fixed, 5-2D-random, 8-2D-fixed, 8-2D-random, 5-3D-fixed, 5-3D-random, 8-3D-fixed, 8-3D-random).

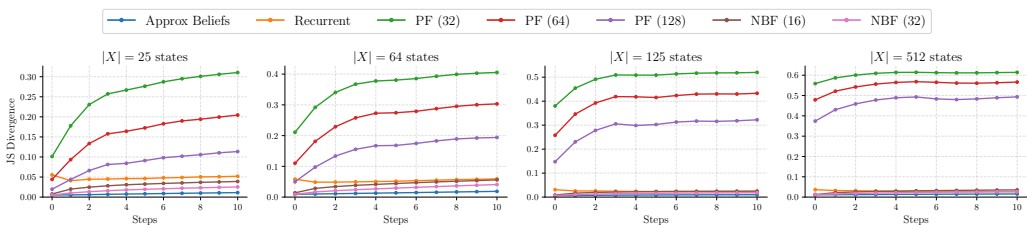

Figure 4: Jensen-Shannon divergence on **fixed** grids and policies (left to right: 5-2D, 8-2D, 5-3D, 8-3D). Training is repeated for 100 random seeds, with each model evaluated over 500 episodes. Shaded areas indicate $\pm 1$ standard error on the average model performance.

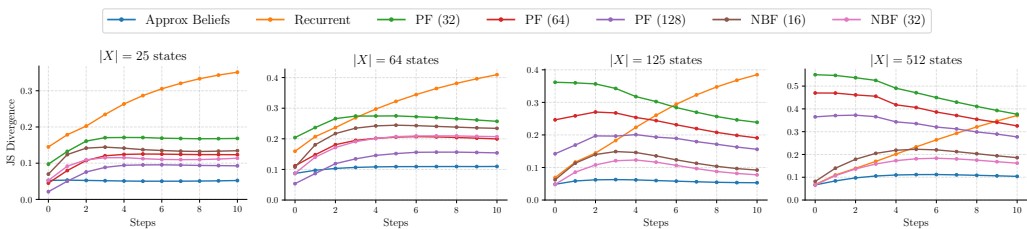

Figure 5: Jensen-Shannon divergence on **randomized** grids and policies (left to right: 5-2D, 8-2D, 5-3D, 8-3D). Training is repeated for 100 random seeds, with each model evaluated over 500 episodes. Shaded areas indicates $\pm 1$ standard error on the average model performance.

**Fixed Grids and Policies.** Figure 4 summarizes results for fixed grids and policies. The belief model computes its embedding using samples from the target distribution, so it provides an expected performance ceiling for NBF. This shows that the embedding is expressive enough to model the set of belief distributions in this partially observable fixed grid. The recurrent approach is capable of modeling posterior updates on a fixed grid, and further tuning could potentially allow it to perform better than the belief model in the fixed setting. NBF maintains a low JS divergence, comparable to the belief model over many steps, while using a relatively low number of particles. This suggests that NBF's update is effective for approximating the posterior computation in the embedding space. Even with orders of magnitude more particles than NBF, the PFs struggle to achieve comparable performance and demonstrate scalability issues with grid size and dimensionality.

**Randomized Grids and Policies.** Performance in randomized grids and policies is summarized in Figure 5. Belief embeddings effectively model this much larger set of belief distributions (given that the policy and obstacle placement are now randomized and changing at every episode). With no ability to incorporate policy and grid information into the model, the performance of the Recurrent filter degrades significantly compared to the fixed grid setting. This highlights its limited adaptability to non-stationary environments, regardless of its capacity to express complex belief states in fixed settings. Particle-based methods are more robust to dynamic grids and policies, with NBF performing the best overall despite using relatively few particles. NBF's performance gain over particle filtering likely arises because the belief-embedding model captures relevant information about $\mathcal{P}_G^\Pi$.

## 5.2 PARTIALLY-OBSERVABLE GOOFSPIEL

Our second set of experiments uses a modified version of the card game Goofspiel (Lanctot et al., 2013) with $k \in \{4, 5, 6, 7\}$ cards. This domain is a standard benchmark in imperfect information games, and provides a concrete example of when an acting agent must consider the opponent policy for belief computation. $k$-card Goofspiel performance is summarized in Figure 6. Modeling late-game belief states in Goofspiel seems more challenging than in Gridworld. We see this in the growing error of the "Approx Beliefs" filter as the size of the game increases. This may be due to strong constraints on *legal* states (e.g. hand sizes when $t$ cards have been played). Unsurprisingly, significant inaccuracies in the embedding model appear to have negative downstream effects on NBF's performance. On the other hand, the particle filter's performance improves at later timesteps

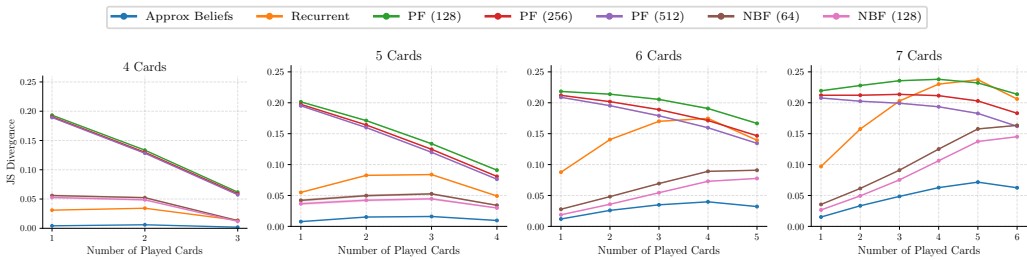

Figure 6: Jensen-Shannon divergence on partially-observable Goofspiel with four, five, six, and seven cards. Training is repeated for 5 random seeds, with each model evaluated over 500 episodes for 10 random seeds. Shaded areas indicate $\pm\,1$ standard error on the average model performance.

as belief state entropy drops. Despite these difficulties, NBF still outperforms the particle filters with an order of magnitude fewer particles (64 vs. 512) in all four sizes.

## 5.3 Triangulation

Finally, we evaluate NBF in a domain with a continuous state space we call *Triangulation*, where the agent must localize itself in $\mathbb{R}^2$ by moving and scanning fixed beacons before stopping as close as possible to the origin. Full details are available in the supplementary material.

Exact posterior computation is not feasible in this domain, so we use a large particle filter (1024 particles) as the ground truth for both training the belief model and computing the JS divergence for evaluation. JS divergence is computed by discretizing the state space into a grid and calculating the probability of each cell according to the filters. Performance in Triangulation is summarized in Table 1. We observe that NBF with only 16 particles performs substantially better than the particle filter baselines.

Table 1: Jensen-Shannon Divergence on Triangulation. Training is repeated for 20 random seeds, with each model evaluated over 100 episodes.

| PF (32) | PF (64) | PF (128) | PF (256) | NBF (16) | NBF (32) |
|---|---|---|---|---|---|
| $0.638 \pm 0.001$ | $0.607 \pm 0.001$ | $0.565 \pm 0.001$ | $0.513 \pm 0.001$ | $0.459 \pm 0.002$ | $0.459 \pm 0.002$ |

## 6 Discussion

Though our empirical evaluation focuses on three relatively simple domains, it still highlights NBF's versatility and potential effectiveness in more complex tasks. For deep recurrent approaches to modeling $g(\theta, y)$, scalability and model expressiveness are insignificant when they cannot incorporate critical environment information $(G, \pi)$. Constraining the particle budget to grow sub-linearly with $|X|$ immediately exposes the scaling pathology of classical particle filters, even in our smallest testbeds. NBF achieves good performance with orders-of-magnitude fewer particles, whereas PFs remain inaccurate despite far larger particle sets. In such cases, the additional cost of embedding and generating a much smaller set of particles with our model is insignificant compared to particle simulation costs. Confirming this in larger domains for downstream tasks such as learning and sequential decision-making is a promising avenue for future work.

In light of our promising results, NBF has limitations related to its belief embedding model and particle-based updates. For instance, experiments on Goofspiel highlighted the importance of an accurate belief embedding model. In some cases, filtering performance could be highly dependent on choosing appropriate task-specific architectures and training methods.

Training data for the domains tested in this work is both easy to generate and reflective of the set of belief states encountered during filtering. Learning an embedding model from the data encountered while filtering would make NBF applicable to settings where representative training data is

difficult or impossible to obtain before filtering. That said, many state-of-the-art online search algorithms (Silver et al., 2017; Moravčík et al., 2017; Schrittwieser et al., 2020; Schmid et al., 2023) require significant computation for offline training but keep online search at decision-time less computationally intensive. These settings suit NBF perfectly and match the experiments conducted in this paper. While NBF's posterior updates reduce the chance of particle impoverishment, they do not eliminate it, especially under extreme conditions where impoverishment can occur in a single step. Such updates can also potentially incur computational overhead during inference compared to pure model-based approaches or the analytical updates of some classical filtering methods. In this sense, NBF trades inference speed for increased representational capacity and adaptability to environmental and control dynamics.

### 6.1 RELATED WORK

There is a notable connection between variational hidden states in deep recurrent models and belief state modeling (Chung et al., 2015). Recurrent neural filtering algorithms (Krishnan et al., 2015; Karl et al., 2016; Lim et al., 2020; Revach et al., 2022) can incorporate external observations and learn the overall transition dynamics defined with a fixed $\pi$ and $G$. However, this implies that, regardless of their expressivity, these models are fixed at test time and cannot easily be adapted to new transition dynamics. Alternative methods (Fickinger et al., 2021; Sokota et al., 2022) *fine-tune* a large pre-trained deep generative model via gradient updates at decision time to adapt to changes in the environment dynamics. The model is initially trained on a large sample set aggregated from many belief states and then refined to fit a test-time belief state. Like NBF, such methods can resample from the full support of the distribution, which mitigates impoverishment risks. However, this comes at a cost as fine-tuning may require many costly gradient updates for each target belief state. This makes it less suitable as a component of fast online search algorithms.

Alternative approaches for embedding beliefs map distributions into an RKHS via kernel mean and conditional mean embeddings (Song et al., 2009). This yields unique representations under characteristic kernels. However, tying representation quality to a fixed kernel risks mismatch with the posterior family encountered at test time, and unlike NBF, these embeddings are not naturally generative.

Belief state modeling has often been implicitly studied in downstream tasks such as search and learning in partially observable environments. The aforementioned fine-tuning approaches (Fickinger et al., 2021; Sokota et al., 2022) have been applied to search and learning in Hanabi. POMCP (Silver & Veness, 2010) performs Monte Carlo Tree Search from particle-based approximations of belief states. Neural Filtering and Belief Embedding can potentially act as a drop-in replacement for particle filtering and offer richer belief state approximations for search. Likewise, Šustr et al. (2021) uses particle-based approximations of value functions for depth-limited search. Approximating value functions in the embedding space is also a promising avenue for future work.

## 7 CONCLUSION

We introduced Neural Bayesian Filtering, a method for modeling belief states in partially observable Markov systems. It models the set of distributions induced by a Markov system as a latent space and performs particle-based posterior updates in this latent space upon new observations. Its underlying models for embedding beliefs are trained strictly from sample sets of example belief states, and its posterior update directly integrates non-stationary dynamics and control variables. We show empirically in three partially observable domains that it retains the robustness of traditional particle filtering while approximating rich, multimodal belief states with far fewer particles. Neural Bayesian Filtering has potential applicability well beyond the tasks demonstrated in this paper, extending naturally to various domains involving sequential decision-making, planning, and estimation under uncertainty.

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

## A  NBF Pseudocode

We present the full pseudocode for Neural Bayesian Filtering. The algorithm updates the input belief embedding by generating and then simulating $n$ particles for a single step. These particles are weighed according to the probability of input observation $y$ given the environment dynamics $G$ and control variable $\pi$.

---

**Algorithm 1:** Neural Bayesian Filtering

---

**input** : $\theta$ — Belief Embedding, $y$ — Observation, $G$ — Environment, $\pi$ — Control,
$(\psi, \phi)$ — Model Parameters, $n$ — Number of Particles, $\varphi$ — function to estimate $\mathbb{E}_p[\varphi]$
**output:** $\theta'$ — Updated Belief Embedding

1  $z_{1:n} \overset{\text{iid}}{\sim} \mathcal{N}(0_d, I_d)$      // $n$ samples of $d$-dimensional Gaussian noise
2  $x_{1:n} \leftarrow f_\psi(z_{1:n}; \theta)$      // Generate particles from belief embedding
3  **for** $i \leftarrow 1 \dots n$ **do**
4       $x'_i \sim T_G(x_i, \pi)$
5       $w_i \leftarrow H_G(x_i, x'_i)[y]$
6  **end**
7  $\hat{\mu}^{(n)}(\varphi) \leftarrow \frac{\sum_{i=1}^n w_i \varphi(x_i)}{\sum_{i=1}^n w_i}$      // Estimate target expectation
8  **return** $\mathcal{E}_\phi(x'_{1:n}, \texttt{normalize}(w_{1:n})), \hat{\mu}^{(n)}(\varphi)$      // Output embedding and estimate $\mathbb{E}_p[\varphi]$

---

The algorithm can optionally estimate the expectation (Line 7) of a target function $\varphi$ over the belief state. Next, we show theoretical results about the convergence of the estimate when NBF has access to a perfect belief embedding model.

## B  Proof of Theorem 4.1

The theorem requires two main assumptions: a perfect belief embedding model and $\epsilon$-global observation positivity. A *perfect* embedding model has of two properties. First, for every $p(x) \in \mathcal{P}_G^\Pi$, there exists a $\theta_p^* \in \mathbb{R}^m$ such that $p_{\theta_p^*}(x) = p(x)$. Second, for any finite collection of samples $x_{1:n} \in X^n$ and weights $w_{1:n} \in \mathbb{R}^n$, $\sum_{i=1}^n w_i = 1$, if the weighted empirical distribution induced by $(x_{1:n}, w_{1:n})$, $\hat{p}(x)$ is equal to $p(x)$, then $\mathcal{E}_\phi(x_{1:n}, w_{1:n}) = \theta_p^*$.

Define the successor pairs of state space $X$ given $(G, \pi)$ as $\text{succ}_{G,\pi}(X) \overset{\text{def}}{=} \{(x, x') : x, x' \in X, T_G(x, \pi)[x'] > 0\}$. $\epsilon$-*global observation positivity* of $(G, \pi)$ states that there exists an $\epsilon \in (0, 1]$ such that for all $y \in Y$ and $(x, x') \in \text{succ}_{G,\pi}(X)$:

$$T(x, \pi)[x'] \cdot H_G(x, x')[y] \geq \epsilon$$

This means that for $G$ and $\pi$, every transition has some probability of generating any observation $y$. In practice, a small $\epsilon$ is sufficient to guarantee NBF's consistency. Without this simplifying assumption, there can be a non-zero (but vanishing) chance that the sample weights in the estimator $\hat{\mu}^{(n)}(\varphi)$ are all equal to zero. In a practical setting where this happens, one could repeat lines 2-6 of Algorithm 1 until some $w_i > 0$. With finite state and observation spaces and these assumptions, we can prove the almost-sure convergence of NBF to the target posterior. First, we provide a lemma for the strong law of large numbers for self-normalizing importance samplers. The proof is an adaptation of the one found in Owen (2013).

**Lemma B.1** (Strong Law for Self-Normalized Importance Sampling Estimators). *Given a finite sample space $X$, let $p(x)$ be a target distribution and $q(x)$ be a proposal distribution such that*

$q(x) > 0$ *whenever* $p(x) > 0$. *Let* $W : X \to (0,1]$ *be a weight function such that* $p(x) = c \cdot W(x)q(x)$ *for normalization constant* $c$. *Finally, let* $\varphi : X \to \mathbb{R}$ *be any bounded function.*

*Draw samples* $x_1, \ldots, x_n \overset{iid}{\sim} q$. *Let* $w_i = W(x_i)$ *and*

$$\hat{\mu}^{(n)}(\varphi) = \frac{\sum_{i=1}^n w_i \varphi(x_i)}{\sum_{i=1}^n w_i}$$

*be the self-normalized importance sampling estimate of* $\mathbb{E}_p[\varphi]$. *Then,*

$$\hat{\mu}^{(n)}(\varphi) \xrightarrow{a.s.} \mathbb{E}_p[\varphi]$$

*as* $n \to \infty$.

*Proof.* Since the pairs $(w_i, x_i)$ are i.i.d., the numerator $S_n \overset{\text{def}}{=} \sum_{i=1}^n w_i \varphi(x_i)$ and denominator $B_n \overset{\text{def}}{=} \sum_{i=1}^n w_i$ of the estimate are i.i.d sums. With bounded $\varphi$ and $W$, we can apply Kolmogorov's Strong Law of Large Numbers to $S_n$ and $B_n$, which gives

$$\frac{S_n}{n} \xrightarrow{a.s.} \mathbb{E}_q[\varphi W], \qquad \frac{B_n}{n} \xrightarrow{a.s.} \mathbb{E}_q[W]$$

as $n \to \infty$.

By the definition of $q$ and $W$, $1 = \sum_x p(x) = \sum_x cW(x)q(x) = c\mathbb{E}_q[W]$. Thus, $\mathbb{E}_q[W] = c^{-1}$. Since $\mathbb{E}_q[\varphi W] = c^{-1}\mathbb{E}_q[\varphi W c] = c^{-1}\mathbb{E}_p[\varphi]$.

Thus, by the Continuous Mapping Theorem (Mann & Wald, 1943), we have

$$\hat{\mu}^{(n)}(\varphi) = \frac{S_n/n}{B_n/n} \xrightarrow{a.s.} \frac{c^{-1}\mathbb{E}_p[\varphi]}{c^{-1}} = \mathbb{E}_p[\varphi]$$

as $n \to \infty$. $\qquad\qquad\qquad\qquad\qquad\qquad\qquad\qquad\qquad\qquad\qquad\qquad\quad \square$

Applying this Lemma to the one-step particle update of NBF lets us show that if the current estimate $\theta$ is consistent, then $\theta$ is also consistent in the limit.

**Theorem B.2** (NBF Consistency). *Assume $\epsilon$-global observation positivity of $(G, \pi)$ and a finite $X$ and $Y$. For any finite horizon $t_{max}$, belief state $p_t(x), t \leq t_{max}$, and any bounded function $\varphi : X \to \mathbb{R}$, let*

$$\hat{\mu}_t^{(n)}(\varphi) = \frac{\sum_{i=1}^n w_i \varphi(x_i)}{\sum_{i=1}^n w_i}$$

*be the estimate of* $\mathbb{E}_{p_t}[\varphi]$ *computed by NBF with a perfect embedding model and $n$ particles. Then,*

$$\sup_{0 \leq t \leq t_{max}} |\hat{\mu}_t^{(n)}(\varphi) - \mathbb{E}_{p_t}[\varphi]| \xrightarrow{a.s.} 0$$

*as* $n \to \infty$.

*Proof.* According to Algorithm 1, the weight for transitioning from $x_i$ to $x_i'$ is $w_i = T_G(x_i, \pi)[x_i'] \cdot H_G(x_i, x_i')[y]$. We start by proving almost sure convergence for any fixed $t \leq t_{\max} : t, t_{\max} \in \mathbb{N}$ by induction.

**Base case.** $t = 0$    Here $p(x) = p_0$, so a perfect embedding model implies we can compute $\theta_p^*$ by embedding samples from $p_0$ and then generate $x_{1:n} \overset{iid}{\sim} p(x)$. $\varphi$ is bounded, so we can apply the Strong Law of Large Numbers to get the result.

**Inductive step.**    Assume for some $t < t_{\max}$, generated particles $x_{1:n}$ are i.i.d. according to $p_t(x)$.

After the loop in Algorithm 1, we have $x_{1:n}'$ distributed according to the proposal $q(x) = \sum_{x'} p_t(x')T_G(x', \pi)[x]$ with weight function $W = H_G(x', x)[y]$. Note that the exact posterior at time $t + 1$ can be written as

$$p_{t+1}(x) = \frac{\sum_{x'} p_t(x') T_G(x', \pi)[x] \cdot H_G(x', x)[y]}{c} = q(x)W(x)/c \tag{1}$$

for some normalization constant $c$.

$\epsilon$-global observation positivity implies that for all $x \in X$ both $p_{t+1}(x) > 0 \implies q(x) > 0$ and $W(x) > 0$ (see Algorithm 1, Line 5). Since $T_G$ and $H_G$ output probability mass functions, $W(x) \leq 1$ for all $x \in X$. As a result, we can apply Lemma B.1 and get that $\hat{\mu}_{t+1}^{(n)}(\varphi) \xrightarrow{\text{a.s.}} \mathbb{E}_{p_{t+1}}[f]$ as $n \to \infty$. This implies that embedding $\theta_{p_{t+1}}^* = \mathcal{E}_\phi(x'_{1:n}, w_{1:n})$ and regenerating $x''_{1:n} \overset{\text{iid}}{\sim} p_{\theta_{t+1}^*}(x) = p_{t+1}(x)$ using a perfect model gives the desired result.

**Almost sure convergence of the sequence.** Now that we have shown almost sure convergence for any $t \leq t_{\max}$, we can complete the proof of the theorem. For any $\delta > 0$, there exists a random integer $N_t = \min\{n : \forall m \geq n, |\hat{\mu}_t^{(n)}(\varphi) - \mathbb{E}_{p_t}[\varphi]| \leq \delta\}$. Let $N = \sup_{0 \leq t \leq t_{\max}} \{N_t : t \leq t_{\max}\}$, then for all $n \geq N$ we have $|\hat{\mu}_t^{(n)}(\varphi) - \mathbb{E}_{p_t}[\varphi]| \leq \delta$ for every $t \leq t_{\max}$.

$\square$

**Corollary B.3.** *Under the same conditions as Theorem 4.1, as $n \to \infty$,*

$$\sup_{0 \leq t \leq t_{max}} |\hat{\mu}_t^{(n)}(\varphi) - \mathbb{E}_{p_t}[\varphi]| = O_p(n^{-1/2})$$

*Proof.* From Equation 1, let $\tilde{p}_t(x) \overset{\text{def}}{=} cp_t(x) = W(x)q(x)$ and denote $w_i \overset{\text{def}}{=} W(x_i) = \frac{\tilde{p}_t(x_i)}{q(x_i)}$ for $1 \leq i \leq n$. Let $Z_i \overset{\text{def}}{=} w_i(\varphi(x) - \mathbb{E}_p[\varphi])$. Then,

$$\mathbb{E}_q[Z_i] = \sum_x q(x)w_i(\varphi(x) - \mathbb{E}_p[\varphi])$$
$$= \sum_x \tilde{p}_t(\varphi(x) - \mathbb{E}_p[\varphi])$$
$$= c\mathbb{E}_p[\varphi] - c\mathbb{E}_p[\varphi] = 0$$

So $Z_i$ have mean zero and are independent.

Now take the denominator of the estimate $B_n = \sum_{i=1}^n w_i$. By $\epsilon$-global observation positivity, $w_i \geq \epsilon$, so $B_n \geq n\epsilon \implies B_n^{-2} \leq (\epsilon^2 n^2)^{-1}$.

Since,
$$\hat{\mu}_t^{(n)}(\varphi) - \mathbb{E}_{p_t}[\varphi] = \frac{\sum_{i=1}^n Z_i}{B_n},$$

it follows that, by independence and zero mean of $Z_i$,

$$\mathbb{E}[(\hat{\mu}_t^{(n)}(\varphi) - \mathbb{E}_{p_t}[\varphi])^2] = \mathbb{E}[B_n^{-2}(\sum_{i=1}^n Z_i)^2] \leq \frac{1}{\epsilon^2 n^2} \sum_{i=1}^n \text{Var}[Z_i].$$

$\varphi$ is bounded, so $\text{Var}[Z_i] \leq \text{Var}_{p_t}[\varphi] \leq ||\varphi||_\infty^2 < \infty$. Define $\sigma_t^2 \overset{\text{def}}{=} \text{Var}_{p_t}[\varphi]$, so $\sum_{i=1}^n \text{Var}[Z_i] \leq n\sigma^2$. Plugging this into the previous bound and taking the supremum over $t$ gives

$$\sup_{0 \leq t \leq t_{\max}} \mathbb{E}[(\hat{\mu}_t^{(n)}(\varphi) - \mathbb{E}_{p_t}[\varphi])^2] \leq \sup_{0 \leq t \leq t_{\max}} \frac{\sigma_t^2}{\epsilon^2 n} = \frac{\sigma^2}{\epsilon^2 n}$$

for $\sigma^2 \overset{\text{def}}{=} \sup_{0 \leq t \leq t_{\max}} \sigma_t^2$.

Applying Chebyshev's inequality gives, for any $\delta > 0$

$$\Pr[|\hat{\mu}_t^{(n)}(\varphi) - \mathbb{E}_{p_t}[\varphi]| \geq \delta] \leq \frac{\sigma_t^2}{\epsilon^2 n \delta^2}$$

Therefore,

$$\Pr[\sup_{0 \le t \le t_{\max}} |\hat{\mu}_t^{(n)}(\varphi) - \mathbb{E}_{p_t}[\varphi]| \ge \delta] \le \frac{(t_{\max} + 1)\sigma^2}{\epsilon^2 n \delta^2}$$

Setting $\delta = M/\sqrt{n}$ for $M > 0$ gives:

$$\Pr[\sqrt{n} \sup_{0 \le t \le t_{\max}} |\hat{\mu}_t^{(n)}(\varphi) - \mathbb{E}_{p_t}[\varphi]| \ge M] \le \frac{(t_{\max} + 1)\sigma^2}{\epsilon^2 M^2}$$

which lets us apply the definition of stochastic boundedness (Van der Vaart, 2000) to finish the proof. □

## C  WALL-CLOCK TIME EXPERIMENTS

We conducted experiments comparing the time required to perform update steps in the filtering algorithms from Section 5. The experiments were restricted to fixed two- and three-dimensional grids of sizes 5 and 8. Since differences in other environments arise only from the transition functions, which are identical across all filters, we did not include them. We benchmarked the runtime of a single update step using `time.perf_counter()` on a 2024 MacBook Pro with a 12-core M4 Pro processor. The final results, reported in milliseconds, are presented in Table 2. We report the mean and standard deviation computed from 10,000 update-step measurements. To mitigate the influence of outliers, we applied IQR filtering, removing any measurements outside the range $[Q1 - 1.5 \times \text{IQR}, ; Q3 + 1.5 \times \text{IQR}]$, where the interquartile range is defined as $\text{IQR} = Q3 - Q1$.

The results in Table 2 show that the cost of inference in NBF can be comparable to a particle filter with more particles. This depends on model size, number of particles, and the complexity of simulating the environment one step. Complex environments may require more expressive, slower models, but on the other hand, computation time may also be dominated by particle simulation.

Table 2: Time (in milliseconds) needed to perform one update step during filtering.

|          | 5-2D                  | 5-3D                  | 8-2D                  | 8-3D                  |
|----------|-----------------------|-----------------------|-----------------------|-----------------------|
| Recurrent | $0.1714 \pm 0.0042$  | $0.1731 \pm 0.0043$   | $0.1718 \pm 0.0044$   | $0.1737 \pm 0.0047$   |
| PF (32)  | $0.2931 \pm 0.0074$   | $0.3094 \pm 0.0068$   | $0.2961 \pm 0.0068$   | $0.3243 \pm 0.0109$   |
| PF (64)  | $0.4074 \pm 0.0134$   | $0.4325 \pm 0.0129$   | $0.4223 \pm 0.0126$   | $0.4436 \pm 0.0148$   |
| PF (128) | $0.7191 \pm 0.0246$   | $0.7588 \pm 0.0238$   | $0.7464 \pm 0.0259$   | $0.7736 \pm 0.0279$   |
| NBF (16) | $0.5939 \pm 0.0129$   | $0.6273 \pm 0.0078$   | $0.6053 \pm 0.0078$   | $0.6295 \pm 0.0075$   |
| NBF (32) | $0.8243 \pm 0.0184$   | $0.7952 \pm 0.0080$   | $0.8350 \pm 0.0143$   | $0.7955 \pm 0.0079$   |

To compare with methods that rely on gradient fine-tuning, we also measured the wall-clock time of gradient updates for the recurrent model used in the Gridworld experiments in Section 5. The reported results (Table 3) show the average time needed for one update step, using a precomputed batch of data of size 32. Batch computation is excluded from the timing, and the average is computed over 10 000 gradient update steps.

The results in Table 3 show that even a single gradient update on a relatively small recurrent network takes significantly more time than an update step in either of the two filters. Though the increased cost of a single update step may seem acceptable given the favorable time needed to perform one filtering update step (one forward pass), test-time gradient fine-tuning may require hundreds or thousands of gradient updates (Sokota et al., 2022).

Table 3: Time (in milliseconds) needed to perform one gradient update of a Recurrent filter.

|          | 5-2D                  | 5-3D                  | 8-2D                  | 8-3D                  |
|----------|-----------------------|-----------------------|-----------------------|-----------------------|
| Recurrent | $2.3056 \pm 0.0230$  | $2.9947 \pm 0.0445$   | $2.6437 \pm 0.0518$   | $5.1424 \pm 0.1303$   |

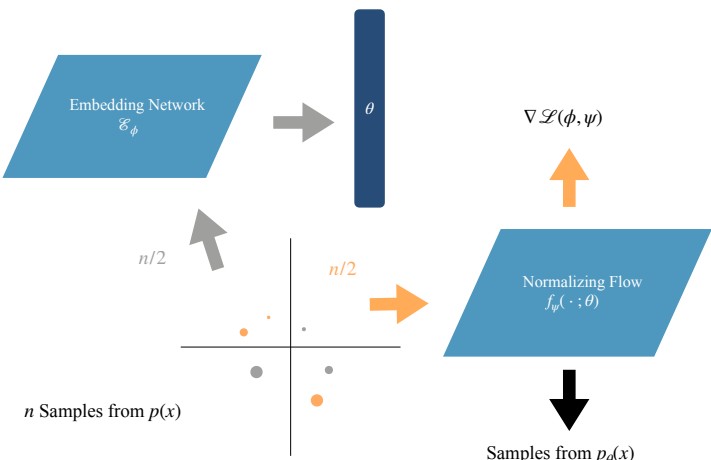

Figure 7: Training belief embedding models. Colored arrows show the flow of training sample points from the target distribution.

# D    ADDITIONAL EXPERIMENTAL DETAILS

Figure D outlines the training process for the belief models used to embed distributions in all of our experiments. Belief embedding models are trained by generating $n$ samples from a belief state instance $p(x)$ from the set of target distributions $\mathcal{P}_G^\Pi$. Half of the samples are used to compute the embedding $\theta$ that conditions the generative model $f_\psi(z; \theta)$. The rest of the samples are used to approximate the gradient of $\mathcal{L}(\phi, \psi)$.

All experiments were implemented in Jax using standard libraries from the Jax ecosystem. We used custom implementations for Normalizing Flows and all three environments. All source code is available as part of the supplementary material. Next, we provide domain-specific details about the models used in our experiments.

## D.1    GRIDWORLD

In our Gridworld experiments, the observer has access to the agent's policy and a simulator for the grid. Policies are generated by biasing the agent's movement toward a randomly selected goal—softmax temperature controls policy entropy to create noise in the agent's path. In each configuration, the number and size of obstacles are constant, but their location is either fixed or randomized. Each experiment was repeated for 500 episodes to compute a model's average JS divergence at a given step, and model training was repeated for 100 random seeds. Parameters for the set of grids used for training and evaluation are shown in Table 5.

**Belief Embedding Model.**    The embedding function $\mathcal{E}_\phi$ consisted of a standard MLP with 3 layers of 128 units each and ReLU activations. The generative model $f_\psi(z; \theta)$ used a uniform prior and 5 coupling layers (Dinh et al., 2016) with masked inputs. Variational dequantization (Ho et al., 2019) was performed to smooth discrete grid locations. More hyperparameters are shown in Table 4.

**Recurrent Model.**    The recurrent baseline learns a mapping from a sequence of observations about movement in the grid to an approximation of $p(x)$. These observations are the same as those used to define the posteriors in the filtering tasks. Since belief states in Gridworlds of these sizes are small, the model outputs a softmax distribution over potential grid locations. JS divergence between model output and belief state instances was minimized directly. More hyperparameters are shown in Table 4.

**Computational Resources.**    For each random seed, model training required roughly 2 CPU-hours on commodity consumer hardware. Each evaluation (consisting of 500 episodes) took at most 3

CPU-hours. Since every experiment was repeated for 100 seeds, the end-to-end compute requirements were roughly 500 CPU-hours for each of the 8 grid configurations. We used cluster resources provided by a source that will be revealed upon publication.

Table 4: Gridworld model and training hyperparameters.

|  | Belief Embedding Model | Recurrent Baseline |
|---|---|---|
| Embedding size | 32 | – |
| Embedding network hidden units | 128 | – |
| Embedding network hidden layers | 3 | – |
| Dequantization hidden units | 32 | – |
| Dequantization hidden layers | 2 | – |
| Normalizing Flow / RNN hidden units | 32 | 32 |
| Normalizing Flow / RNN hidden layers | 5 | 2 |
| Normalizing Flow coupling layers | 5 | – |
| Batch size | 32 | 32 |
| Training steps | 100 000 | 100 000 |
| Training samples (per $p(x)$) | 64 | – |
| Optimizer | AdaGrad | AdaGrad |
| Learning rate | 0.10 | 0.10 |

Table 5: GridWorld environment parameters.

| Parameter | $5 \times 5$ | $8 \times 8$ |
|---|---|---|
| Obstacle cubes | 1 | 2 |
| Cube width | 2 | 3 |
| Softmax temp. (for random policies) | $1 \times 10^{-5}$ | $1 \times 10^{-5}$ |

## D.2 GOOFSPIEL

In $k$-card Goofspiel, both players and the prize deck start with the same set of cards, labeled 0 through $k - 1$. A round starts when a *prize card* is revealed, indicating the value of winning the round. Players act by simultaneously *bidding* a card and then observe only the outcome of who played the highest card (win, draw, or loss). In our variant, the card symmetry is broken: each player and the prize deck receives a random subset of size $k - 1$, while all other rules remain unchanged. Small $k$ means exact posterior computation is tractable, enabling efficient training and evaluation of our models and baselines.

During training, samples are obtained by following policies of both players to a randomly selected depth and sampling opponent action histories from the true posterior given the generated observations. The policies are sampled randomly from a pool generated by independent self-play using PPO (Schulman et al., 2017). These policies were randomly split into a training and test set used only for evaluation. We trained each model on five different random seeds and each filter's reported performance is averaged over 10 different runs, each consisting of 500 episodes.

**Goofspiel Policy Generation.** We generated a sequence of policies by independent self-play using PPO to simulate the effect of changing policies during learning, as in classical self-play settings. We used the Jax version of StableBaselines 3 (SBX) (Raffin et al., 2021)—modified to support action masking. In self-play, we trained a policy against its previous checkpoint for 524 288 timesteps and saved a checkpoint every 4096 timesteps. We repeated this self-play loop four times, producing a sequence of 512 policies in total.

**Belief Embedding Model.** The belief embedding model for Goofspiel uses a Standard Normal prior, and consists of a variational dequantization layer parameterized by a single coupling layer,

followed by a series of coupling layers. The dequantization coupling layer uses an affine transformation, and each of the following layers use one-dimensional non-linear squared (NLSq) transformations. All coupling layers transform masked inputs. After dequantization, each coupling layer is further parameterized by $\theta$. Concrete hyperparameters used in our experiments are listed in Table 6.

**Recurrent Model.** Observations in Goofspiel consist of features such as the player's hand, the prize deck, the current one-hot encoded prize card, and the winnings. The recurrent baseline learns a mapping from a sequence of these observations to an embedding. This embedding conditions a Normalizing Flow with the same architecture as described above. The key difference is that the recurrent model maps the observation sequence to an embedding directly, whereas the belief embedding model embeds sample sets from $p(x)$.

**Computational Resources.** Goofspiel experiments were run on computing resources provided by a source that will be revealed upon publication. For each game size and each random seed, belief model training required approximately 12 hours and recurrent model training approximately 3 hours on 32 CPU cores. Afterwards, each model was evaluated for 500 episodes, which took between several minutes and three hours on 12 CPU cores, depending on the size of the game. The evaluation was repeated 10 times.

Table 6: Goofspiel model and training hyperparameters.

|  | Belief Embedding Model | Recurrent Baseline |
|---|---|---|
| Embedding size | 48 | 48 |
| Embedding network LSTM hidden units | – | 64 |
| Embedding network LSTM hidden layers | – | 2 |
| Embedding network MLP hidden units | 128 | 64 |
| Embedding network MLP hidden layers | 3 | 2 |
| Dequantization hidden units | 48 | 48 |
| Dequantization hidden layers | 2 | 2 |
| Normalizing Flow MLP hidden units | 128 | 64 |
| Normalizing Flow MLP hidden layers | 4 | 4 |
| Normalizing Flow coupling layers | 8 | 8 |
| Batch size | 64 | 64 |
| Training steps | 150 000 | 16 000 |
| Training samples (per $p(x)$) | 64 | 32 |
| Optimizer | Nadam | Nadam |
| Learning rate | 0.001 | 0.001 |

## D.3 TRIANGULATION

Triangulation is a noisy localization task on a bounded grid. At the start of each episode, the agent is placed uniformly at random in $[-5; 5]^2$. Every timestep, it can move 0.5 units in any of the four cardinal directions, issue a `stop` action to end the episode, or `scan` to query a range sensor. The objective is to navigate as close as possible to the origin and then `stop`. Performance is evaluated by measuring the Euclidean distance to the origin at termination. Both motion execution and range measurements are corrupted by Gaussian noise. Beacons are fixed and located at $(-2, -2)$, $(0, \sqrt{8})$, and $(2, -2)$. Only one beacon is "active" at any time step, and the active identity rotates deterministically each step in a fixed cyclic order. A `scan` action returns a noisy scalar equal to the distance from the agent's current (noisy) position to the currently active beacon. This induces partial observability even with frequent scans: the agent must reason jointly about its position and the active-beacon phase while trading off movement toward the origin with information-gathering scans, under both transition and measurement noise.

The set of policies used to control agents in our experiments consists of mixtures between single cardinal actions and `scan`. The policy for each episode is chosen at random and is available to the filters at evaluation time.

**Belief Embedding Model.**     The belief embedding model for Triangulation uses a Standard Normal prior and consists of a series of coupling layers that use parameterized affine transformations and transform masked inputs. No dequantization is necessary because the domain is continuous, and each coupling layer is further parameterized by $\theta$. Hyperparameters were not tuned extensively, and are listed in Table 7.

**Computational Resources.**     Triangulation experiments were run on computing resources provided by a source that will be revealed upon publication. A model for this environment can be trained in a few minutes on a single GPU.

Table 7: Triangulation model and training hyperparameters.

|  | **Belief Embedding Model** |
| --- | --- |
| Embedding size | 32 |
| Embedding network MLP hidden units | 128 |
| Embedding network MLP hidden layers | 3 |
| Normalizing Flow MLP hidden units | 64 |
| Normalizing Flow MLP hidden layers | 2 |
| Normalizing Flow coupling layers | 6 |
| Batch size | 32 |
| Training steps | 30 000 |
| Optimizer | Adam |
| Learning rate | 0.001 |

### D.4   DONUTS

For our illustrative example from Section 3, we used a scaled-down version of the belief embedding model used in the main experiments. Model and training hyperparameters are shown in Table 8. Donuts models train in several minutes on a laptop.

Table 8: Donuts model and training hyperparameters.

|  | **Normalizing Flow Model** | **Cond. FM Model** |
| --- | --- | --- |
| Embedding size | 8 | 8 |
| Embedding network hidden units | 64 | 64 |
| Embedding network hidden layers | 3 | 3 |
| Dequantization hidden units | – | – |
| Dequantization hidden layers | – | – |
| Normalizing Flow MLP hidden units | 32 | 64 |
| Normalizing Flow MLP hidden layers | 3 | 4 |
| Normalizing Flow coupling layers | 8 | – |
| Batch size | 32 | 32 |
| Training steps | 30 000 | 30 000 |
| Training samples (per $p(x)$) | 128 | 128 |
| Optimizer | Adam | Adam |
| Learning rate | 0.001 | 0.001 |

## E   ILLUSTRATIVE EXAMPLE USING CONDITIONAL FLOW MATCHING

To highlight the versatility of our proposed approach, we applied a Conditional Flow Matching (CFM) model (Lipman et al., 2022; Tong et al., 2023) to the donut-shaped distributions introduced as a toy domain described in Section 3. We experimented with both independent coupling and

optimal transport (OT) coupling (Lipman et al., 2022) to define the target vector fields that generate the conditional probability paths in our flow matching models. An example of a CFM model trained with optimal transport coupling predicting the density for a randomly sampled set of donuts is shown in Figure 8(b). The model was conditioned using 256 samples sampled from the true distribution and asked to predict the density of each point in a $512 \times 512$ grid. Model and training hyperparameters are shown in Table 8.

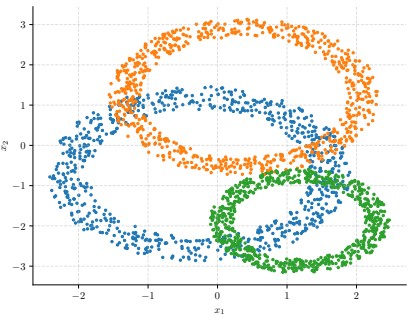
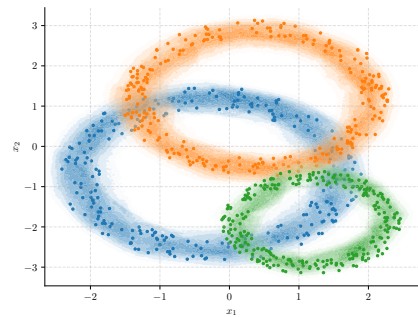

(a) Sample donut distributions. Each distribution has three parameters: mean, radius, and width.

(b) Learned densities after conditioning the model on 256 samples from the target distribution.

Figure 8: Embedding the set of donut distributions in $\mathbb{R}^2$ using Conditional Flow Matching

## F  CODE

We will release all of our training and evaluation code on GitHub upon publication.

