# OpenReview forum: "Neural Bayesian Filtering"
_ICLR.cc/2026/Conference — Submitted to ICLR 2026_

### Official Review · Reviewer_SgMX · 2025-10-20

**Soundness:** 2
**Presentation:** 3
**Contribution:** 3
**Rating:** 4
**Confidence:** 2

**Summary:**

The paper introduces NBF which is a new way to do Bayesian filtering when the system is only partially observable. Instead of tracking the belief state with huge num of particles like a normal particle filter, they learn a compact embedding $\theta$ that represents the whole belief distribution. This embedding is built so it does not depend on the order or number of samples and can capture multi-modal posteriors.

At each step, NBF draws new particles from a generative model conditioned on this embedding, updates them using the transition and observation models, and then re-embeds the results to get the new $\theta$. This avoids the usual particle impoverishment issue and makes filtering more efficient. They test NBF on small partially observable tasks. The results show that NBF can stay close to the true posterior, while using fewer particles than a standard PF.

**Strengths:**

- The paper builds a permutation and size invariant embedding for belief states, so the order or number of particles doesn’t matter. It’s a clean way to represent complex, even multi-modal posteriors as a single latent vector.

- Instead of doing the usual particle-filter resampling that kills diversity, the paper redraw particles from a learned flow model conditioned on the current embedding that keeps the diversity of samples.

- The paper includes a short proof showing that, under reasonable conditions, the embedding update converges to the true posterior.

- Able to handle discrete states cases.

**Weaknesses:**

- The paper use a particle-filter–style update step when incorporating new observations, but it’s not super clear why that specific choice. PFs are flexible and model-free, sure, but they also bring sampling noise and inefficiency. It would be nice to see some ablation or comparison with other update mechanisms to show this choice outperformance. Just example: using a learned parametric update network or simpler probabilistic approximations (e.g. Gaussian or flow-based direct updates) instead of your PF, or using your proposed recurrent model in update process but still keeping $\theta$ embedding. Showing how these variants perform would help justify sticking with the PF-style update.

- The chosen baselines (vanilla particle filter and the Approx Beliefs oracle) are fine but not enough. To really compare this work against the SOTA, it would help to include some of the newer learned filtering methods that also approximate posteriors in high-dimensional or multimodal settings.
For example, recent works like [1],[2] and [3] all solve posterior approximation by generative models but with different parameterizations. Including two or three of them would make the evaluation more complete.

- As mentioned in the discussion, the current experiments are mostly toy setups and simple. Comparing to similar recent papers in top venues, adding at least one higher-dimensional or real-world experiment would strengthen the work.

- The paper lacks an easy-to-follow code setup or README-style instructions. Although it mentions that the code will be released upon acceptance, it would be nice to have an anonymous link or at least some pseudo-code included in the paper to make the method easier to follow.

[1] - Wan, Ziyu, and Lin Zhao. "DiffPF: Differentiable Particle Filtering with Generative Sampling via Conditional Diffusion Models." arXiv preprint arXiv:2507.15716 (2025).

[2] - Chen, Xiongjie, and Yunpeng Li. "Normalizing Flow-Based Differentiable Particle Filters." IEEE Transactions on Signal Processing (2024).

[3] - Younis, Ali, and Erik Sudderth. "Learning to be Smooth: An End-to-End Differentiable Particle Smoother." Advances in Neural Information Processing Systems 37 (2024): 7125-7155.

**Questions:**

Thank you for the draft and all of the efforts. I have some questions regarding the paper:

- Can you justify why this specific update scheme fits NBF best? Why not a few ablation test with other parameterizations to show that PF is indeed the most effective choice?

- what happens if you replace the PF step with a simple parametric or recurrent update but still using the same embedding $\theta$? How would that work, did you this try in your exps? It would help clarify how much of the gain actually comes from the PF structure versus the embedding itself.

- Similar papers in top venues usually include at least one high-dimensional or real-world experiment. How does your model handle such cases, and what modifications would be needed to make it works in higher dimensions? If not adding such an experiment, could you clearly mention these in your discussion?

- The idea of using a multimodal embedding $\theta$ as a kinda universal posterior generator is interesting. But how flexible and accurate is it in practice? I.e. if you train task-specific embedding networks instead of a shared one, do you see a large performance improvement or just a marginal gain? Did you try this in your exps?

- In Figure 4 (25 states), it looks like the particle filter performs even better than the Approx Beliefs baseline in the first few steps, which is a bit unexpected since Approx Beliefs directly uses samples from $p(x)$. Any explanation for this?

---

> ### Author Response · Authors · 2025-12-02
>
> We would like to thank the reviewer for their thorough review. We appreciate all the comments and apologize for not including the code in our initial submission. Although our initial version of the paper includes pseudo-code detailing the inference step of NBF, we will include pseudo-code for training as well. Additionally, we have attached a ZIP archive containing the used libraries and their version, the training and evaluation code, along with a detailed README.md file. We remain committed to open-sourcing the provided code on GitHub upon publication.
>
> We have also included an additional high-dimensional environment that should better demonstrate the scaling capabilities of NBF. Please refer to the Additional Experiments response for further details.
>
> **Particle-filter-style Updates.** The main motivation behind this choice is the assumption of the perfect model of the environment. It is definitely possible and sometimes even needed to learn the transition dynamics, whether explicitly (i.e. model-based RL style) or implicitly (e.g. what our Recurrent baselines does) but in this paper we do not study such a problem. Learning a compact representation of belief states is a hard and unsolved problem by itself and may be regarded as a stepping-stone towards more general methods that would not need to rely on an exact model of the environment.
>
> Interestingly, the Recurrent baseline we used in Goofspiel is somewhat close to your suggestions. The embedding network is an LSTM which receives an observation as its input. The LSTM's hidden state is used to parameterize the same Normalizing flow model as in NBF. However, such an approach has limited adaptability to changing policies and non-stationary environments, and requires fine-tuning to adapt to unseen situations. We have shown experimentally the impact on the performance and the computational requirements in our Gridworld experiments.
>
> **Weak baselines.** Thank you for providing references to Differentiable Particle Filters, we will include them in the Related Work section. We have read through the papers and while they aim to solve an interesting problem, it is different from the problem we solve. NBF assumes the aforementioned availability of a perfect model of the environment and does not try to learn the transition or the observation dynamics in any way.
>
> To the best of our knowledge, there is no work that investigated the same setup where they try to learn a compact representation of the posteriors while utilizing the available environment dynamics. We believe that comparing to Differentiable PFs is unfair as they work with a weaker set of assumptions, and learning the transition or the observation model is naturally a more difficult problem. We will make sure to better delineate the differences and discuss where exactly NBF fits in the context of prior work.
>
> > Similar papers in top venues usually include at least one high-dimensional or real-world experiment. How does your model handle such cases, and what modifications would be needed to make it works in higher dimensions? If not adding such an experiment, could you clearly mention these in your discussion?
>
> We have included a new high-dimensional experiment in the Additional Experiments response. The only limitation that NBF faces when scaling to larger domains is the need for representative training data. This and a sufficient representational capacity of the neural networks is enough to apply NBF to any domain. As mentioned in the General response, lifting this limitation is a natural next step, but it is unfortunately beyond the scope of this paper. We discuss this limitation in Section 6, but we will make sure to communicate this more clearly in the improved version of the paper.
>
> > The idea of using a multimodal embedding  as a kinda universal posterior generator is interesting. But how flexible and accurate is it in practice? I.e. if you train task-specific embedding networks instead of a shared one, do you see a large performance improvement or just a marginal gain? Did you try this in your exps?
>
> Although we agree that such an ablation would be interesting, it is out of scope for this paper. One motivation of NBF is to filter accurately and robsutly in large policy spaces (which are likely to occur during downstream tasks like learning. Training a separate embedding network for each policy is orthogonal to this goal.

---

### Official Review · Reviewer_yyuh · 2025-10-31

**Soundness:** 2
**Presentation:** 1
**Contribution:** 2
**Rating:** 2
**Confidence:** 3

**Summary:**

This work proposes a particle filter (PF) type algorithm which, instead of resampling, maps the set of weighted samples to a latent embedding at each time step, and then "refreshes" the particle system by replacing the weighted particles by a new set of evenly weighted particles generated by a normalising flow parametrised by the embedding. The embedding seems to be trained assuming that IID samples from the filtering distributions are available.

The proposed method seems to me to be a type of "regularised particle filter" (e.g., [1]). Regularised particle filters have long been favoured by some researchers and practitioners as a way of overcoming the particle degeneracy problem. They approximate the weighted sample by some (e.g., continuous) distribution and then sample from this approximation instead of resampling from the original weighted sample.

[1] Musso, C., Oudjane, N., & Le Gland, F. (2001). Improving regularised particle filters. In Sequential Monte Carlo methods in practice (pp. 247-271). New York, NY: Springer New York.

**Strengths:**

Here, the authors use latent embeddings (and normalising flows) for constructing (and sampling from) the regularisation in a regularised particle filter which seems novel to me and could potentially be powerful. Although I am not necessarily following all the related literature. The theoretical results seem correct although they rely on very strong assumptions.

**Weaknesses:**

**Significance.**

The theoretical results (almost sure convergence and $L_p$ error bounds) are likely correct. However, they are not surprising under the very strong assumptions (and more general results under weaker assumptions are well known in the particle filtering literature). I don't think a methodology paper like this necessarily needs extensive theory to justify publication in ICLR. However, I will note that the $L_p$ error bound does not prove stability, i.e., the error bound likely grows with $t$. And it would be crucial to show that the method is stabie (i.e., that error do not accumulate over time) under more realistic assumptions on the embedding.


**Clarity:**

The paper has problems with clarity which are so severe that there are parts of the methodology and numerical results that I cannot follow/verify:

1. There are a large number of undefined or poorly defined symbols and terms: $\Delta X$, $\Delta Y$, $d$, $T_G$, $H_G$, $K$, $N$ (= $n$?). $\phi, $p_\theta^{(k)}$, "cardinality-invariant function".

2. Pseudo-code is only given in the appendix and is difficult to understand because it only seems to cover the case of a single observation (and doesn't cover the training procedure, details on how the weighted samples are mapped to the latent space, nor multiple time steps).

3. The numerical experiments cannot be understood without reading the appendix. There are very likely strengths and weaknesses of the methodology upon which I cannot comment because I was unable to understand the numerical results section for this reason.


**Other:**

1. Code should always be made available to reviewers.

2. There are some inadequate references: citing Papamakrios et al., 2021, for the well known density transform formula on Page 4, and Sokota et al. (2022) for the notion of resampling in particle filters.

**Questions:**

1. What is the computational cost of the different approaches shown in the numerical results and is it comparable (even when taking training into account)?

---

> ### Author Response · Authors · 2025-12-02
>
> We thank the reviewer for the helpful comments and suggestions and will implement them to improve the paper.
>
> **Poorly defined terms.** $\Delta X$ is the set of probability distributions over the sample space $X$. $d$ is used to indicate dimensionality throughout. $T$ and $H$ are clearly defined to represent the transition and emission functions. The $G$ subscript indicates that they are components of the FOSG $G$. $K$ and $N$ are clearly defined on line 195 immediately before their usage in the training objective. $\phi$ is the learnable parameter vector for the embedding function, and $p_\theta(x)$ is defined on line 158. Permutation and cardinality invariance is explained on line 159. Though all of these terms are indeed defined in the paper, we will make sure to make them clearer in the revision.
>
> **Psuedocode / code.** As mentioned in other responses, we are committed to adding pseudocode related to the training procedure and will include the source code for all experiments as part of this discussion phase.
>
> **Uninterpretable numerical results.** The numerical results simply show the Jensen-Shannon divergence of each algorithm's approximation with the ground truth beliefs. As we mention in the paper, a lower divergence indicates a higher fidelity approximation. Could the reviewer elaborate on what is unclear about the results?
>
> **Computation cost.** Inference-time costs are measured in the appendix (Table 2) and show NBF has little to no overhead compared to the baselines. Training time varies per task, but data generation is typically the bottleneck. Since the models are relatively small, we train all those used in this paper in under 1 hour on a commodity laptop.

---

### Official Review · Reviewer_Q1E7 · 2025-10-31

**Soundness:** 3
**Presentation:** 3
**Contribution:** 3
**Rating:** 4
**Confidence:** 3

**Summary:**

This work presents an algorithm for learning the embedded representations of belief states in partially observable environments combined with generative modelling for efficient Bayesian updates. The authors define an embedding function, which approximates features of a target distributiong. They prove that the NBF estimate almost surely converges to the true distribution under certain conditions, and demonstrate the viability of their method in the deep setting in 3 benchmarks.

**Strengths:**

1. The necessity of learned belief states is well motivated, given the limited expressivity of traditional methods.
2. The updating of the NBF by posterior sampling is well-explained.
3. Theorems 4.1 and 4.2 are well-explained, and Figure 3 further exemplifies the elegance of the proposed method.
4. The proof of convergence of NBF with a perfect model is convincing, and provides a basis for its viability in practical settings (although the gap between this and a practical model is not quantified).
5. The improvement in divergence between the true distributions in all 3 tasks with NBF is convincing - even with fewer particles, NBF is able to clearly approximate beliefs on gridworlds, discrete POMDP Goofspiel, and the triangulation game better than particle filtering.

**Weaknesses:**

1. The donut distribution example in Figure 2 is necessary but not sufficient; why was it not shown the method would also work on an arbitrary and visualisable 2D distribution? This would demonstrate the usefulness of the proposed method over standard Gaussian models.
2. The examples provided remain toy domains with relatively low dimensions, and do not necessarily demonstrate the viability of NBF in a larger, more random, or non-stationary POMDP setting such as [1]. This is currently the main drawback of this work, and a study of NBF vs particle filtering in a larger partially-observable setting would likely suffice a recommendation for acceptance.
3. Lack of ablations;  a study disentangling the contribution of the embedding model from the filtering update is crucial in understanding whether gains arise from better density modelling, resampling, or something else.
4. (Minor) While the success of NBF in belief estimation is well-demonstrated, it remains to be seen whether this has a significant effect on the performance, robustness, or generalization of a system which uses those beliefs is proportionally improved. While it may not be well within the scope of this paper, a study on the performance of a reinforcement learning algorithm with the belief states for policy learning would be interesting.

[1] Tao, Ruo Yu, et al. "Benchmarking Partial Observability in Reinforcement Learning with a Suite of Memory-Improvable Domains." arXiv preprint arXiv:2508.00046 (2025).

**Questions:**

1. The belief state space in the gridworld and triangulation settings is clearly defined, but is not apparent in Goofspiel. What is it?

---

> ### Author Response · Authors · 2025-12-02
>
> Thank you for your review.
>
> **The Donuts Domain.** We would like to clarify the motivation behind including the illustrative example which we call Donuts. We included this domain simply for the purposes of providing visual intuition of what the embeddings represent and how they can be leveraged to reconstruct distributions from finite sets of samples. It is merely meant as a means of introducing the reader to the idea of embedding distributions only from sets of their samples, nothing more. The experiments in Section 5 (plus the additional response period experiments) are meant to demonstrate the effectiveness of NBF in learning complex multi-modal distributions reliably.
>
> **Toy Examples.** We have created a new domain that allows us to obtain representative high-dimensional training data needed for NBF's training. We have included the description of the experiment and its preliminary results in the Additional Experiments response. We commit to including a complete set of experiments on this domain in the full version of the paper. Please refer to the Additional Experiments response for further information.
>
> **Lack of Ablations.** We thank the reviewer for this comment. Further ablations studying how embedding size and other modeling choices impact the performance of NBF, and we commit to adding them to the appendix. We believe our current experimental results show that NBF is robust and performs well with relatively simple models of modest computational complexity.
>
> **Downstream Task Performance.** We fully agree that it remains to be seen whether the use of learned embedding will translate to improved performance in downstream tasks, such as game playing. We believe that to be out of scope for this paper and leave it to future work.
>
> > The belief state space in the gridworld and triangulation settings is clearly defined, but is not apparent in Goofspiel. What is it?
>
> The belief space in Goofspiel is the distribution over the opponent's valid hands. In Imperfect Information Goofspiel, we only observe whether we won or not, but not the card the opponent played. Either outcome may leave multiple possible cards the opponent could have played, which in turn determines the possible hands the opponent can have.

---

### Official Review · Reviewer_MSmT · 2025-11-02

**Soundness:** 3
**Presentation:** 2
**Contribution:** 3
**Rating:** 6
**Confidence:** 2

**Summary:**

This paper proposes a new method for belief state modeling by approximating the posterior distribution with a generative model conditioned on belief state embeddings. The proposed method is validated in variants of Gridworld, the card game Goofspiel, and a continuous localization environment called Triangulation, and benchmarked agains other classical methods such as particle filtering and the recurrent approach.

**Strengths:**

- The goal of overcoming particle impoverishment in high-dimensions is well-motivated.

- The proposed framework to combine classical filtering, conditional embeddings and deep generative models is novel.


- Empirical studies show that NBF consistently outperforms particle filtering.

**Weaknesses:**

- The presentation of the method can be improved. A few important details are not clear and it would be helpful to have a complete algorithm for the proposed method (see more details in Questions section). And the context provided in  the preamble of sect. 3, particularly regarding $\pi$ and $G$ is not referenced in the remaining of the section.

- The clarity of the remaining part of the paper can be improved. See more detailed comments in the Questions section.

- It is hard to assess the significance of the proposed method, given the limited tasks and baseline methods.

**Questions:**

Method related:

- Can the authors discuss practical choices of embedding networks?

- Is there a particular reason to use normalizing flows for the conditional distribution fitting part?

- Does the embedding network require pre-training before applying it to decision making tasks?

- Do the parameters of embedding network and the normalizing flows get updated during filtering?

- What are the takeaways from Sec 3.2?

- What are the criteria for a good belief embedding model?



Empirics related:

- Can the authors explain what is the recurrent approach? Is there a particular implementation / reference  for that?

- In table 1, why not include the comparison to the recurrent approach?

- How representative and challenging are the benchmarks problems in this filed? As the authors pointed out in the Discussion, "Though our empirical evaluation focuses on three relatively simple domains, it still highlights NBF’s versatility and potential effectiveness in more complex tasks.", why not consider more complex tasks?

- How many tasks are truly multimodal? Is there empirical evidence to illustrate that the proposed method captures the multimodality, if any?

- Is there similar weight collapse issue in combining the embedding for the proposed method?

---

> ### Author Response · Authors · 2025-12-02
>
> We thank you for dedicating your time to help us improve our paper. We have carefully gone through all of your and other reviewers' comments and questions. We hope the following responses to your questions are satisfactory.
>
> We understand and appreciate the suggestions for improving the presentation of NBF.
> We commit to implementing them and including more details about the experiments, the algorithm (including more detailed pseudo-code), baselines, and environments.
> We have also include an additional experiment that better demonstrates NBF's scaling capability to high-dimensional environments.
>
> > Can the authors discuss practical choices of embedding networks?
>
> NBF requires the embedding network to be permutation and cardinality-invariant. Aside from that, many options may be viable. We have mainly experimented with two types of the embedding network: 1) a version that applies the same MLP to each particle independently and then mean pools the resulting embeddings, and 2) the DeepSets architecture. We saw no tangible benefit empirically from the more complex aggregation found in DeepSets.
>
> > Is there a particular reason to use normalizing flows for the conditional distribution fitting part?
>
> The choice of generative model is also somewhat arbitrary. We included a small demonstration of using Conditional Flow Matching on the Donuts domain in Appendix E. Generally, other types of models (e.g. Flow Matching or Diffusion) could be used within NBF.
>
> > Does the embedding network require pre-training before applying it to decision making tasks?
>
> Yes, we assume the ability to generate trajectories from a representative set of policies and environments to train the embedding network.
>
> > Do the parameters of embedding network and the normalizing flows get updated during filtering?
>
> No, they do not. NBF consists of two separate phases -- the training and the inference (filtering) phase. During the training phase, the networks are trained to match the samples from a dataset consisting of sets of samples, with each set representing a single distribution. In the filtering phase, the networks remain frozen.
>
> > What are the takeaways from Sec 3.2?
>
> Section 3.2 serves as an illustrative example of the training process and provides a visualization of the ground truth (Figure 2a) and how the networks are able to recover the distributions from sets of 64 samples (Figure 2b). It also provides a visual intuition of what the embeddings are able to represent.
>
> > What are the criteria for a good belief embedding model?
>
> A good embedding model accurately represents the set of target belief distributions in the task and produces a permutation and cardinality invariant embedding given samples from the belief distribution.
>  We have done very little tuning and hyperparameter search before reporting the final results in the paper. A study of what constitutes a good belief embedding model and what tradeoffs various generative models bring is, unfortunately, beyond the scope of this paper.
>
> > Can the authors explain what is the recurrent approach? Is there a particular implementation / reference for that?
>
> The recurrent approach in our experiments is an LSTM network, which receives an observation on the input.
> It is primarily included, not as a true baseline, but to show the consequences of trying to learn the non-stationary transition dynamics in the system.
>
> > In table 1, why not include the comparison to the recurrent approach?
>
> We commit to including the Recurrent baseline in the Triangulation environment.
>
> > How representative and challenging are the benchmarks problems in this filed? As the authors pointed out in the Discussion, "...", why not consider more complex tasks?
>
> As this is a concern shared by multiple reviewers, we have responded to this issue in the general response and we have now included preliminary results on a high-dimensional environment, which are available in the Additional Experiments response.
>
> > How many tasks are truly multimodal? Is there empirical evidence to illustrate that the proposed method captures the multimodality, if any?
>
> All three tasks in the original paper are multimodal. The multimodality in Gridworld can be seen in Figure 1. In Imperfect Information Goofspiel, we only get to observe whether we won or not, leaving many possibilities what card the opponent could have played, which in turn leads to many possible clusters of opponent's hands. Lastly, Triangulation is multimodal by design.
> Another example comes from poker, where an opponent's action history and policy could indicate they have either a high pair or a flush with high probability.
> Downstream reasoning in this case would require taking the expectation over both modes.
>
> > Is there similar weight collapse issue in combining the embedding for the proposed method?
>
> The traditional weight collapse over many steps that PFs suffer from is less likely as NBF propagates the particles only for a single step.

---

### Author Response · Authors · 2025-11-18
**General Response to Reviewers**

We would like to sincerely thank the reviewers for their reviews and their time spent to help improve our work. We appreciate all the comments and will integrate them into a revised version of the paper.

All four reviewers have raised many valid points, some of which will require more time to adequately respond to. We will post our initial replies to each reviewer in the coming days. However, in the meantime, to make the most of the discussion period, we wanted to respond to common issues raised by several of the reviewers as soon as possible to facilitate further discussion. We believe this will help us improve the paper even more.

First of all, two of the reviewers have pointed to the missing code. We apologize for not including the code in our initial submission. We are working on cleaning and anonymizing the codebase, and we will upload it as part of the supplementary materials within the next couple of days. Furthermore, we maintain our commitment to open-sourcing our code on GitHub upon publication.

The overall theme among the reviewers is that the chosen environments are low-dimensional and do not demonstrate how NBF would scale to larger domains well enough. The main limitation of NBF currently lies in its need for a representative offline training dataset. We are aware of this limitation and discuss it in Section 6. Lifting this limitation would make NBF more applicable and allow filtering in domains where it was previously impossible to do efficiently, including many of the benchmark POMDP problems mentioned by Reviewer Q1E7. We feel that solving this problem is out of scope for this paper and is a promising future research direction.

The rationale behind the choices of domains we made is two-fold: 1) the need for representative training data required domains where the exact computations of the posterior distributions are feasible, and 2) the posteriors had to be complex enough to demonstrate the weaknesses of traditional filtering algorithms, such as the Kalman and Particle filters. We commit to including an additional high-dimensional domain where the aforementioned filters may work, but which will better demonstrate the scaling capabilities of NBF.

The second point that has been raised repeatedly is the choice of particle-filter-style updates of the distribution embeddings. Unlike other works, such as those proposed by Reviewer SgMX, that propose differentiable particle filters, we do not learn the transition dynamics or the observation function of the environment. Instead, we focus on the variant of the problem where those functions are available. Part of our motivation is that NBF allows us to adjust to non-stationarity in the transition dynamics caused by, for example, changes to the policy.

Lastly, some of the reviewers mentioned issues with the clarity of the presentation. We commit to addressing these problems, including inadequate references and undefined mathematical symbols and terms. We will also improve the description of NBF by adding pseudo-code detailing the training process, as well as a better description of belief states in each of the environments.

---

### Author Response · Authors · 2025-12-02
**Additional Experiments**

We provide an additional set of experiments and their preliminary results on a continuous high-dimensional domain, which we call Mixture of Gaussians (MoG). In our original experiments, we used environments that are complex enough to pose a significant challenge to Particle Filters but are low-dimensional and allow for closed-form solutions to their beliefs. We have included an additional domain that allows for arbitrary dimensionality but can be solved via Kalman filtering. Kalman filtering allows us to track the posterior distributions needed to obtain training data.

MoG consists of a randomly initialized mixture of Gaussians. MoG($n$, $k$) is parameterized by the number of dimensions $n$ and the number of components in the mixture $k$. Each component has its own mean vector, covariance matrix, and transition and observation dynamics. At each step, the components are simulated using their respective transition dynamics, and one randomly chosen component is used to produce the observation that the filters receive. A single episode lasts for 10 steps.

The increased dimensionality prevents us from using the Jensen–Shannon divergence as a performance metric. Therefore, we report the results on MoG using Maximum Mean Discrepancy (MMD). MMD is widely used in generative modeling research. It only requires samples and is therefore more suitable for high-dimensional distributions than JS divergence. We use the RBF kernel with $\sigma = \sqrt{n}$.

The following are preliminary results on two configurations -- MoG(50, 16) and MoG(75, 16). We trained five different models for 100 000 steps; each was evaluated for 100 episodes and the results were averaged. The following are the mean MMD ($\pm$ std. error) values averaged over five random seeds. We can see that even in high-dimensional continuous problems, NBF outperformsß Particle filters with significantly fewer particles.
We commit to adding a full description of the environments and the training and evaluation setup including the code upon publication.

|                   |  PF (32)      |         PF (64)        |      PF (128)        |      PF (256)         |     NBF (16)        |      NBF (32) |
-------------|---------------------|---------------------|---------------------|---------------------|--------------------- |---------------------
  MoG(50, 16)  |  1.3369 $\pm$ 0.0150  | 1.3156 $\pm$ 0.0123  | 1.3020 $\pm$ 0.0119  | 1.2899 $\pm$ 0.0124  | 1.1975 $\pm$ 0.0190 |  1.1898 $\pm$ 0.0193 |
  MoG(75, 16)  |   1.3530 $\pm$ 0.0104 |   1.3403 $\pm$ 0.0077 |  1.3300 $\pm$ 0.0083  | 1.3194 $\pm$ 0.0081 |  1.0662 $\pm$ 0.0067 |   1.0528 $\pm$ 0.0071 |

---

### Meta-Review · Area_Chair_9pmb · 2026-01-07

**Summary:**

Reviewers were generally unified in their evaluation that the work focuses on solving an important problem of modeling complex, multimodal belief distributions using a Bayesian approach. However, there were serious concerns about the clarity of the presentation, including the description of the method itself, and the scale of the experiments. The experiments focused on toy settings that do not adequately show the ability of this approach to scale beyond other belief state methods. Reviewers also mentioned a lack of ablations that made it difficult to detect where performance gains were actually coming from. Finally, there were questions about the choice of method itself -- why use particle filters rather than any other approach, which was not properly ablated in the paper.

**Reviewer Concerns:**

Unfortunately, I believe most of the reviewer concerns are still unaddressed by the rebuttal. The authors have included a high-dimensional experiment, but it is still toy in the sense that it is using generated data of a mixture of Gaussians, rather than anything from the real world, which would be much more convincing as to the expressibility of this approach. The authors were able to explain why they use a particle-based approach, but only by limiting themselves to specific settings. The authors were not able to improve the clarity of their exposition during the rebuttal phase or include ablations or control experiments.

**Reviewer Scores:**

I believe reviewer SgMX may have increased their score due to the authors addressing their question about why they used particle-based filters rather than differentiable approaches, although the author response was somewhat weak in that they propose to focus on settings where the transition dynamics or the observation function from the environment are provided. However, most other concerns were not adequately addressed by the rebuttal response so I believe no other reviewers would have increased their score.

---

### Decision · Program_Chairs · 2026-01-26

Reject